# Regulation of Actin Filament Length by Muscle Isoforms of Tropomyosin and Cofilin

**DOI:** 10.3390/ijms21124285

**Published:** 2020-06-16

**Authors:** Katarzyna Robaszkiewicz, Małgorzata Śliwinska, Joanna Moraczewska

**Affiliations:** Department of Biochemistry and Cell Biology, Faculty of Biological Sciences, Kazimierz Wielki University in Bydgoszcz, 85-671 Bydgoszcz, Poland; robkat@ukw.edu.pl (K.R.); gosia.sl@ukw.edu.pl (M.S.)

**Keywords:** tropomyosin, isoform, cofilin, actin, thin filament

## Abstract

In striated muscle the extent of the overlap between actin and myosin filaments contributes to the development of force. In slow twitch muscle fibers actin filaments are longer than in fast twitch fibers, but the mechanism which determines this difference is not well understood. We hypothesized that tropomyosin isoforms Tpm1.1 and Tpm3.12, the actin regulatory proteins, which are specific respectively for fast and slow muscle fibers, differently stabilize actin filaments and regulate severing of the filaments by cofilin-2. Using in vitro assays, we showed that Tpm3.12 bound to F-actin with almost 2-fold higher apparent binding constant (K_app_) than Tpm1.1. Cofilin2 reduced K_app_ of both tropomyosin isoforms. In the presence of Tpm1.1 and Tpm3.12 the filaments were longer than unregulated F-actin by 25% and 40%, respectively. None of the tropomyosins affected the affinity of cofilin-2 for F-actin, but according to the linear lattice model both isoforms increased cofilin-2 binding to an isolated site and reduced binding cooperativity. The filaments decorated with Tpm1.1 and Tpm3.12 were severed by cofilin-2 more often than unregulated filaments, but depolymerization of the severed filaments was inhibited. The stabilization of the filaments by Tpm3.12 was more efficient, which can be attributed to lower dynamics of Tpm3.12 binding to actin.

## 1. Introduction

Skeletal muscle is a heterogeneous tissue composed of various fiber types, which based on the expression of myosin heavy chain are classified as slow twitch (type I), fast twitch (type IIA, IIX, IIB), and hybrid fibers [1,2]. Another protein which diverged into fast and slow isoforms is tropomyosin, an elongated coiled-coil protein, which extends along seven actin subunits and due to end-to-end interactions forms uninterrupted chains along both sides of the actin filament. In striated muscle, together with troponin complex, tropomyosin controls interactions of actin with myosin heads in a Ca^2+^-dependent way [3]. In skeletal muscle, the contractile thin filaments are associated with three isoforms of tropomyosin—Tpm1.1, Tpm2.2, and Tpm3.12, encoded by *TPM1*, *TPM2*, and *TPM3* genes, respectively. In different species Tpm1.1 and Tpm2.2 were detected in all types of muscle fibers, though at various levels. Although Tpm1.1 and Tpm3.12 are very similar in sequence, Tpm3.12 is expressed exclusively in slow muscle fibers [4,5]. This suggests that Tpm1.1 and Tpm3.12 have different regulatory properties that allow for contraction typical of slow and fast skeletal muscle fibers.

Experimental data strongly suggest that the high sequence similarity of Tpm1.1 and Tpm3.12 is not a consequence of evolutionary redundancy, but is important for specific tropomyosin functions. For example, when Tpm3.12 was overexpressed in the hearts of transgenic mice, it had a hyperdynamic effect on systolic and diastolic function and decreased sensitivity to Ca^2+^ in force generation [6]. Recent studies demonstrated the importance of Tpm1.1 and Tpm3.12 in differentiation of the interactions of fast and slow myosin isoforms with the thin filament. Using in vitro motility assay, a method which allows to analyze the motility of fluorescently-labeled actin filaments gliding on the myosin-coated surface, Matyushenko and colleagues observed that, as compared to Tpm1.1, Tpm3.12 reduced the speed of actin motility over fast and slow myosin by 40–50% [7]. Therefore, not only myosin, but also tropomyosin isoforms contribute to the contractile characteristics of fast and slow muscle fibers.

Numerous myopathy-related mutations were found in *TPM1* and *TPM3* genes, which encode Tpm1.1 and Tpm3.12 isoforms, respectively, however none of the mutations found in Tpm3.12 were identified in Tpm1.1 [8]. The fact that single substitutions in *TPM3* may cause very severe forms of skeletal myopathy characterized by muscle weakness, hypotrophy of type 1 fibers and disproportion in the number of type 1 and type 2 fibers [9,10,11,12], provides additional evidence for the importance of the fiber type-specific tropomyosin isoforms in the function of skeletal muscle.

Fiber type-specific isoforms of myosin heavy chain determine the kinetics of actin-activated ATPase activity, maximal velocity of shortening, force, and tension [13,14,15,16,17]. Force generation can also be affected by the degree of the overlap between actin and myosin filaments [18]. The lengths of thick filaments in slow and fast twitch fibers are very similar, but thin filaments in slow twitch fibers are significantly longer than in fast twitch fibers [19,20]. Thus, the regulation of actin filament length can be an important factor determining the contraction of different types of muscle fibers. In striated muscle, length and stability of thin filaments is maintained by exchange of subunits at the pointed ends, which is controlled by several proteins, including tropomodulin, leiomodin, and cofilin [21,22,23,24].

In mature muscle, the maintenance of the actin filaments’ length and overall organization of the contractile apparatus require cofilin-2, a protein which belongs to the family of actin-depolymerizing factor (ADF/cofilin). The essential role of cofilin-2 in the maintenance of sarcomeric structure after birth was demonstrated in cofilin-2-deficient mice, which suffered from progressive disruption of sarcomere structure with accumulations of F-actin aggregates [25,26]. In sarcomeres, cofilin-2 is enriched at the M-band region, close to the pointed ends of the thin filaments. The depletion of cofilin-2 in cardiomyocytes by RNAi disrupted the periodic pattern and regular lengths of actin filaments [24].

Cofilin-2 is expressed primarily in mature skeletal and cardiac muscle [27,28]. Cofilin binds to monomeric and filamentous actin and promotes actin dynamics by severing and depolymerization of the filaments [29,30]. Cofilin cooperatively binds to F-actin and alters conformation of the filaments [31], resulting in increased fragmentation at the boundaries between cofilin-decorated (cofilactin) and bare filaments [32].

The presence of tropomyosin on the filament was shown to decrease cofilin-induced depolymerization of the filament from the pointed end, but tropomyosin isoforms differ quantitatively in their abilities to regulate cofilin-2 activity, which allows for fine tuning of actin filament dynamics [33,34]. However, it is not known whether tropomyosin isoforms Tpm1.1 and Tpm3.12 differ in the regulation of actin filament dynamics. The goal of this work was to verify the hypothesis that Tpm1.1 and Tpm3.12 differentially stabilize actin and regulate cofilin-2-dependent severing and depolymerization of the filaments. We found that Tpm3.12 promoted longer filaments than Tpm1.1 both in the absence and presence of cofilin-2. The differences between Tpm1.1 and Tpm3.12 in maintaining the filament length can be ascribed to different binding affinities and dynamics of the isoforms.

## 2. Results

### 2.1. Binding of Human Tpm1.1 and Tpm3.12 to F-Actin

Interactions of human tropomyosin isoforms Tpm1.1 and Tpm3.12, specific for fast and slow skeletal muscle fibers, with F-actin were analyzed in vitro by using a co-sedimentation assay. Both isoforms were recombinant proteins carrying Ala-Ser N-terminal extension, which compensated for the lack of the N-terminal acetylation, typical for proteins expressed in bacterial cells [35]. Binding of both isoforms to actin was analyzed before, but under different salt concentrations [36,37]. As tropomyosin interactions with actin are sensitive to the ionic strength [38], this analysis allowed us to determine the concentration of tropomyosin required to saturate actin filaments under conditions, which were close to physiological.

In order to obtain the apparent affinity constants, the experimental points illustrated in Figure 1 were fit to the Hill equation Equation (1). The analysis showed that Tpm3.12 bound to F-actin with almost 2-fold higher affinity than Tpm1.1, but the Hill cooperativity coefficients α^H^ were similar (Table 1). This confirmed the differences in affinities of human Tpm1.1 and Tpm3.12 for F-actin obtained at higher salt concentrations [37,39].

### 2.2. Regulation of Cofilin-2 Binding to F-Actin by Tpm1.1 and Tpm3.12

Next, we analyzed the effects of Tpm1.1 and Tpm3.12 on cofilin-2 binding to F-actin. Using co-sedimentation assay we compared binding of cofilin-2 to F-actin alone with its binding to F-actin saturated with either Tpm1.1 or Tpm3.12 (Figure 2). In agreement with the previous studies [28,33,40], in the absence of tropomyosin cofilin-2 bound strongly to actin filament, but neither Tpm1.1 nor Tpm3.12 affected the saturation of the filament with cofilin-2 (Figure 2A). The apparent association constants obtained by fitting the experimental points to the Hill equation Equation (1) are collected in Table 2. The data show that when the filament was initially saturated with either Tpm1.1. or Tpm3.12, cofilin-2 tended to bind with higher affinity. However, the differences in K_app_ in the presence and absence of tropomyosins were statistically non-significant (*p* > 0.05).

As binding of cofilin along the actin polymer is a cooperative process, it can be considered in terms of a linear lattice model [41]. To fit the experimental points to the McGhee and von Hippel equation Equation (2), the data were mathematically converted. The obtained Scatchard plot is concave downward, which indicates positive cooperativity (Figure 2B). The binding curves fit to the experimental points approach the abscissa at cofilin-2 to actin molar ratio close to 1.0. This shows the maximal occupancy of actin subunit with cofilin-2 and agrees with the known 1:1 cofilin-actin binding stoichiometry [41]. The computed affinity for an isolated binding site (K_0_), and a cooperativity coefficient (ω), which reflects the increase of binding affinity when the ligand binds next to the ligand already bound [42] are collected in Table 2. The binding parameters show that cofilin-2 bound to an isolated site with very low affinity, which was compensated by a high cooperativity coefficient. When cofilin-2 bound to F-actin regulated by tropomyosin isoforms, the affinity for an isolated site increased about 20-fold with a concomitant decrease in the cooperativity. The experimental data mathematically converted to enable the analysis according to the McGhee and von Hippel model are dispersed, because they are sensitive to the experimental error. This yields values of K_0_ and ω that are not accurate, which is why the differences only illustrate the direction of changes in the parameters of the cofilin-2 binding in the absence and presence of tropomyosin isoforms.

### 2.3. Effect of Cofilin-2 on Interactions of Tropomyosin Isoforms with F-Actin

Binding of cofilin-2 to actin was shown previously to remove tropomyosin from the filament, however isoforms dissociated with different efficiencies [33]. To examine the dissociation of Tpm1.1 and Tpm3.12 from the filament by cofilin-2, we used a co-sedimentation assay. Changes in the fractional occupancy of the filament by tropomyosin were analyzed as a function of increasing cofilin-2 concentrations. The effectiveness of cofilin-induced dissociation of tropomyosin can be defined as molar ratio of cofilin to actin required for removal of 50% of tropomyosin from the filament. This parameter, as well as the cooperativity of tropomyosin dissociation, were calculated from the curves obtained by fitting the experimental points to Equation (3). As illustrated in Figure 3A, Tpm1.1 and Tpm3.12 were removed from the filament by cofilin-2 with similar efficiency and cooperativity. The differences in the ratio of cofilin-2 per actin required for half dissociation of both tropomyosins were statistically not significant (Table 3).

To check whether at low occupancy cofilin-2 affects parameters of tropomyosin binding to actin, we examined the association of tropomyosin isoforms to filaments partially saturated with cofilin-2. For that, 1 µM cofilin-2 was used. As shown in Figure 2A (inset), at this concentration cofilin was bound to about 25% of actin filaments. At this concentration of cofilin-2 (0.33 cofilin-2/actin molar ratio) the filament was saturated with tropomyosin in about 80% (Figure 3A). Binding curves illustrated in Figure 3B and the binding parameters collected in Table 1 show that the presence of cofilin-2 reduced the affinity of Tpm1.1 and Tpm3.12 to F-actin. Cofilin-2 had a stronger inhibitory effect on Tpm1.1 affinity than on Tpm3.12. In addition, cofilin-2 exerted different effects on binding cooperativities of tropomyosins (α^H^ in Table 1).

To check whether increasing concentrations of tropomyosin had an effect on cofilin binding to actin, proteins collected in pellets and separated on SDS-gels (Figure 3C) were analyzed quantitatively by densitometry of the protein bands. The normalized densitometric ratio of cofilin or tropomyosin to actin drawn as a function of total tropomyosin concentration illustrates changes in the occupancy of the filament (Figure 3D). The analysis shows that increasing concentrations of tropomyosin did not remove cofilin-2 from the filament, which agrees with the lack of significant effect of tropomyosin on cofilin-2 affinity for F-actin described above.

### 2.4. Effects of Tpm1.1 and Tpm3.12 on Cofilin-2 Induced Fragmentation and Depolymerization of F-Actin

To investigate the effects of Tpm1.1 and Tpm3.12 and cofilin-2 on the length of the filaments, we used fluorescence microscopy assay described before [34]. The visualization of the filaments allowed us to analyze the number of breaks produced by cofilin-2 and to measure the lengths of the filaments. Severing activity was assessed from the analysis of the number of breaks in the filaments, which were observed before and after the addition of cofilin-2. To determine the efficiency of cofilin-2 to depolymerize actin, we measured the length of the longest fragment that was left after the filament had been severed by cofilin.

The analysis of microscopic images of the unregulated and tropomyosin-coated filaments (Figure 4A) showed that ≈80% of the filaments were unbroken and one break appeared in the remaining fraction of actin filaments (Figure 4B). The exposition of the unregulated actin filaments to cofilin-2 for 60 s led to 1–4 breaks per filament, but about 40% of the unregulated filaments remained unbroken. Surprisingly, in the presence of tropomyosin the fraction of unbroken filaments was smaller than without tropomyosin. Sixty seconds after addition of cofilin-2 to the filaments, which were regulated by Tpm1.1, nearly 50% of filaments had one break. It was in contrast to filaments regulated by Tpm3.12, in which 2–4 breaks were dominating (Figure 4C). All differences between the number of breaks in the presence and absence of tropomyosin isoforms were statistically significant (*p* < 0.05).

The analysis of the actin filaments’ length demonstrated that in the absence of cofilin-2 the average length was 10.1 ± 0.65 µm. In the presence of Tpm1.1 and Tpm3.12 the length increased by about 25% and 40%, respectively (time 0 in Figure 5A). The addition of cofilin-2 shortened the tropomyosin-regulated and unregulated filaments, but in the presence of tropomyosin the fragments remaining after severing were longer than in the absence of tropomyosin (Figure 5A). The protective role of tropomyosin isoforms was quantitated by dividing the average filament length computed at each time point by the average values obtained in the absence of cofilin (Figure 5B), which showed that both studied tropomyosin isoforms slowed down the depolymerization of actin filaments by cofilin-2. In the presence of both tropomyosin isoforms the rate of the filament shortening was similar. All differences between the length of filaments obtained in the presence and absence of tropomyosin isoforms and between the filaments decorated with Tpm1.1 and Tpm3.12 were statistically significant (*p* < 0.05).

### 2.5. Dynamics of Tropomyosin Binding to F-Actin

To decipher the mechanism which underlies the differences between tropomyosin isoforms in their ability to diversify the length of the filament we tested the dynamics of tropomyosin binding to actin. To this end we used susceptibility of tropomyosin to proteolysis with trypsin. F-actin, which strongly binds Mg^2+^ (F-Mg-actin) is resistant to digestion with trypsin, except for the last two amino acid residues [43]. As skeletal tropomyosin is sensitive to trypsin digestion [44], we examined the proteolysis time course of Tpm1.1 and Tpm3.12 bound to F-Mg-actin. We assumed that increased dynamics of tropomyosin binding causing more frequent dissociation of tropomyosin molecules from the filament would increase proteolysis.

As shown in Figure 6A, free Tpm1.1 and Tpm3.12 were easily digested by trypsin within 3 min, but in complex with F-Mg-actin they were partially protected, with Tpm3.12 being more resistant to trypsin than Tpm1.1. The quantitative densitometry of tropomyosin bands showed that Tpm1.1 disappeared within 10 min, whereas full digestion of Tpm3.12 required more than 30 min (Figure 6B). During this time the density of F-Mg-actin band was unchanged.

Based on the above results we concluded that Tpm3.12 dissociates from actin less frequently than Tpm1.1, therefore it can stabilize the filament, which increases the length and decreases depolymerization.

## 3. Discussion

In the present work we studied the abilities of Tpm1.1 and Tpm3.12, tropomyosin isoforms expressed in fast and slow muscle fibers, to differentiate lengths of the actin filaments. Such differences would affect the extent of overlap between actin and myosin filaments within sarcomeres of slow and fast muscle fibers and would provide control of the contractile function in addition to the tropomyosin-troponin complex. As muscle isoform of cofilin was shown to participate in the maintenance of the length of sarcomeric actin filaments [24], we checked whether Tpm1.1 and Tpm3.12 diversify actin filament length and control binding, severing and shortening of the filament by cofilin-2. We found that the filaments coated by Tpm3.12 were significantly longer than the filaments covered by Tpm1.1. Both isoforms facilitated severing of the filament induced by cofilin-2, but the resulting fragments were longer than in the absence of tropomyosin.

### 3.1. Structural Determinants of Tpm1.1 and Tpm3.12 Affinity for Actin

The basic features that are crucial for tropomyosin to act as actin regulator are the affinity and cooperativity with which tropomyosin binds to actin. The gene *TPM3* encoding tropomyosin isoform Tpm3.12 evolved from *TPM1* as a result of gene duplication [45]. It can be expected that the effect of amino acid substitutions appearing during molecular evolution is a change in the tropomyosin affinity for actin. Under near native conditions used in this study, Tpm3.12 bound to F-actin with about 2-fold higher affinity than Tpm1.1, but with similar cooperativity. Binding of tropomyosin chains along the filament involves periodic interactions with seven consecutive actin subunits and end-to-end interactions between adjacent tropomyosin dimers [46]. The Hill coefficient suggested that Tpm1.1 bound more cooperatively, however, the high affinity of Tpm3.12 was close to the detection limit of the method, which makes the cooperativity difficult to assess.

A comparison of Tpm1.1 and Tpm3.12 sequences showed that out of 25 amino acid residues which distinguish Tpm1.1 from Tpm3.12, only four are located in consensus actin binding sites within periods 4 and 6. However, there is a strikingly high conservation of amino acid residues located in the consensus actin-binding sites (Figure 7). Only one consensus site residue in Tpm1.1 (Gln135) was changed for Leu in Tpm3.12. This site, however, was shown as less important for tropomyosin interactions with actin [47]. Therefore, there must be other determinants, which account for higher affinity of Tpm3.12 for actin. An interesting possibility is the substitution Arg220Lys, which is located next to two conserved consensus sites—Asp219 and Glu223. Electrostatic repulsion within tropomyosin chain was suggested to set the consensus site residues in a correct orientation for interactions with residues exposed on actin surface [48]. It can be assumed that a substitution of a bulky Arg with smaller Lys may facilitate these intra-chain interactions.

Tpm3.12 carries an extra Met at the N-terminus and Asp2 to Glu3 substitution [8]. These few changes can be responsible for stronger interactions between tropomyosin with actin and between adjacent tropomyosin molecules. Stronger near-neighbor interactions can be particularly important for reduced dissociation of tropomyosin from the filament, which was monitored as lower susceptibility of Tpm3.12 for proteolysis. This in turn can be the key mechanism for decreased dissociation of actin subunits from the ends, which results in longer filaments. High resolution structure of actin filament in complex with Tpm3.12 is still not available, therefore any considerations of the structural determinants of Tpm3.12 interactions with actin are only hypothetical and further experimental verification is necessary.

### 3.2. Tropomyosin-Induced Changes in the Cooperativity of Cofilin-2 Binding to the F-Actin Linear Lattice

Depending on tissue specificity, tropomyosin isoforms have various effects on binding of cofilin to actin filament [24,33,34,40]. In the present and previous study [33] we have demonstrated that skeletal muscle tropomyosin isoforms do not prevent binding of cofilin-2 to actin filament. The saturation of the filament with neither Tpm1.1 nor Tpm3.12 had a significant effect on the affinity of cofilin-2 for actin. Therefore, the affinity of tropomyosin for F-actin does not appear as an important factor, which differentiates binding of cofilin-2 to the filament. Actin filament can be considered as a linear lattice, which interacts with an array of ligands. When bound to the linear lattice, the ligands can be either isolated from other ligands bound to the lattice or can form clusters of contiguously bound molecules, where near-neighbor interactions can affect the ligand affinity through cooperative allosteric effects [42,49]. We found that under conditions used in this study, cofilin-2 bound weakly to the isolated site of bare actin, but with high cooperativity. Such mode of binding is similar to the actin-binding mechanism shown before for non-muscle cofilin-1 [41] and is consistent with structural data showing that cofilin induces conformational changes within actin, which propagates to the neighboring subunits [50,51]. The presence of tropomyosin on the filament changed the way cofilin-2 interacted with actin. Although saturation of the filament with tropomyosin did not affect the overall affinity of cofilin-2 for actin, it increased the affinity for the isolated site and at the same time decreased binding cooperativity.

The reduced cooperativity is expected to limit the size of the cofilin clusters built along the filament and favor shorter but more numerous cofilin-decorated actin segments. As the filament is severed at the boundary between cofilin clusters and cofilin-free subunits [52,53], and the number of cofilins bound in one cluster which are required for effective severing is low [41,54], decreased cooperativity of cofilin binding might increase severing. In the presence of tropomyosin, the number of gaps produced by cofilin-2, which were observed under the fluorescence microscope, was larger than in the absence of tropomyosin. In this regard, the sarcomeric tropomyosins and cofilin-2 can be considered collaborators.

One has to note that single gaps were observed in about 20% of the filaments even in the absence of cofilin-2. Under the experimental conditions used in this work, the severing of the filament could be due to myosin heads (HMM). To immobilize actin, the filaments were bound in rigor (no ATP) to HMM attached to the cover glass, but before the rigor bonds were formed, traces of ATP present in the actin buffer could activate the motor activity of HMM. As HMM was fixed to the surface, cycling cross-bridges could break the filaments.

Due to conformational changes in actin induced by myosin, immobilization of actin filaments with myosin heads can potentially affect the cooperativity of cofilin binding to the filament. However, the high resolution structure of F-actin-Tpm bound to myosin S1 in the rigor state revealed only small alterations in actin structure [55]. Ngo et al. [56] were able to dock cofilin to actin–myosin rigor complex without causing clashes. As in our experiment cofilin-2 was able to sever the filaments, its binding to actin was not severely affected by the myosin heads.

### 3.3. Structural Basis of Tropomyosin Removal from the Filament by Cofilin

Although the presence of tropomyosins on the filament did not prevent binding of cofilin-2, gradual binding of cofilin-2 to the filament removed Tpm1.1 and Tpm3.12 from actin. This seems to be a general phenomenon because it has been observed not only for muscle isoforms (this work, [33,40,57]), but also for non-muscle tropomyosin and cofilin [34]. Under conditions used in this work, cofilin-2 ejected Tpm1.1 and Tpm3.12 with the same effectiveness, which indicates that their binding was not compatible with the structure of cofilactin.

Cofilin binds two actins along one protofilament and induces conformational rearrangement within D-loop located in actin outer domain, which changes the orientation of actin subdomain 2 [54,58]. As skeletal tropomyosin binding to actin is sensitive to the conformation of this region [59], the conformational change in D-loop can be the primary reason of cofilin-induced dissociation of tropomyosin from the filament. In addition, tropomyosin could be released from the filament by the change in shape of the filament, which in cofilactin is more twisted [31]. As actin-binding periods spaced along tropomyosin coiled coil meet charged residues on the face of each actin subunit [60], the cofilin-induced twist of the filament can disrupt the tropomyosin–actin interface and lead to the dissociation of tropomyosin. The effect of cofilin on actin filament structure must be quite dramatic, because a significant reduction of tropomyosin affinity for actin was observed even at low level of actin saturation with cofilin.

### 3.4. Molecular Bases of the Regulation of Actin Filament Length by Tropomyosin Isoforms

Tropomyosin role as actin stabilizing protein, which protects the filaments from depolymerization is well known [61,62]. In the present work we extended this knowledge and demonstrated that muscle isoforms of tropomyosin are able to differentiate length of the filament. In the fluorescence microscopy assay the regulated filaments were significantly longer than the bare F-actin. Both tropomyosin isoforms significantly increased the length of F-actin, but the filaments were longer in the presence of Tpm3.12. A plausible explanation of this difference is that after dilution of F-actin samples, the filaments depolymerized, but the presence of tropomyosin slowed down the depolymerization rate. The second possibility is that the filaments polymerized prior to fluorescence microscopy assay in the presence of tropomyosin were longer than the bare filaments, and after dilution this difference was maintained. When cofilin was added to the flow cell, severing and depolymerization was started, but the length of the remaining filaments was still significantly different. This raises the question of a mechanism that allows the two isoforms of tropomyosin to differentiate the lengths of the filaments.

As judged from the susceptibility to proteolysis, Tpm3.12 dissociates from the filament less frequently than Tpm1.1, hence its higher protective capacity. Despite being partially removed from the filament by cofilin, under conditions used in our fluorescent microscopy assay, both isoforms of tropomyosin protected the filament from cofilin-dependent depolymerization. The mechanism, however, remains elusive.

It is noteworthy that maintaining longer actin filaments is not an inherent feature of all tropomyosin isoforms. In our previous work we have observed that in the presence of non-muscle tropomyosins, which were products of *TPM3* gene (encoding also Tpm3.12), the filaments were shorter than in the presence of products of *TPM1* [34]. Analyses of actin conformational changes associated with binding of different isoforms revealed that tropomyosin isoforms induce different conformational changes in actin regions responsible for stabilization of contacts between actin subunits [63]. Based on these observations, we suggest that tropomyosin-induced conformational changes in actin and different dynamics of tropomyosin binding to actin form the molecular bases of the regulation of actin filament length.

### 3.5. Roles of Tropomyosin Isoforms in the Regulation of Actin Dynamics at the Pointed End

Tropomyosin is not the sole regulatory protein of the thin filament. Except for regulation of the filament on its own, tropomyosin participates in the filament length regulation by tropomodulin. It is possible that Tpm1.1 and Tpm3.12 differentiate the length of the filament indirectly by affecting tropomodulin’s ability to inhibit elongation of the pointed end. Such possibility is suggested by the observations that disease-causing point mutations in Tpm3.12 [37] and Tpm1.1 [64] affect the elongation. Further research is required to fully understand the mechanism of tuning the polymerization at the pointed end by tropomyosin isoforms.

### 3.6. Conclusions

In conclusion, through different stabilization of the thin filaments the muscle-type specific tropomyosin isoforms Tpm1.1 and Tpm3.12 differentiate the length of the thin filaments. Precise regulation of the thin filament length requires concerted action of many proteins, among which tropomyosin isoforms play an important role.

## 4. Materials and Methods

### 4.1. Protein Preparations from Rabbit Muscle

Actin was isolated from rabbit pectoral muscle and purified according to the method described by Spudich [65]. Absorption coefficient 0.63 mg mL^−1^ cm^−1^ at 290 nm and MW 42,000 Da were used to determine the concentration of G-actin.

Heavy meromyosin (HMM) was prepared by TLCK-chymotrypsin (SigmaAldrich, St. Louis, USA) digestion of rabbit fast myosin according to Margossian and Lowey [66]. The concentrations of myosin and heavy meromyosin were determined using absorption coefficients: 0.83 and 0.6 mg mL^−1^ cm^−1^ at 280 nm and proteins MW: 470,000 and 350,000, respectively.

The New Zealand rabbit muscles were a generous gift from the Department of Pathobiochemistry and Clinical Chemistry, Collegium Medicum in Bydgoszcz. All procedures were approved by the Committee for Ethical Experiments on Animals of Collegium Medicum in Bydgoszcz. Permit No 11/2016, granted to dr Eliżbieta Piskorska; valid from 14 April 2016 to 13 April 2021.

### 4.2. Preparation of Recombinant Tropomyosin Isoforms and Cofilin-2

Human Tpm1.1 and Tpm3.12 isoforms were expressed in BL21 (DE3) cells (Novagen Inc, Affiliate of Merck KGaA, Darmstadt, Germany) and purified as described before [37,39]. The sequence of both isoforms was identical with the human Tpm1.1 and Tpm3.12, except for two amino acid extension (Ala-Ser), which is routinely used to compensate for the lack of the N-terminal acetylation in the recombinant proteins expressed in bacterial cells [35]. Protein concentration was determined spectrophotometrically at 280 nm using molar extinction coefficient 17,880. The coefficient was calculated from the human amino acid sequence of Tpm1.1 and Tpm3.12 in the web.expasy.org/protparam/tool.

Cofilin-2 was expressed in BL21 (DE3) cells and purified as described before [33]. The cDNA encoding mouse GST-tagged cofilin-2 subcloned into pGAT2-MM expression plasmid pPL93, was a gift from Pekka Lappalainen, Helsinki University (Helsinki, Finnland). The bacterially expressed protein was purified using GST-affinity chromatography. The GST-tag was removed by digestion with human serum thrombin. The concentration of the protein was determined spectrophotometrically using the absorption coefficient 0.79 mg × mL^−1^ cm^−1^ at 280 nm and MW 18,500 Da.

### 4.3. Fluorescent Labeling of G-Actin with Tetramethylrhodamine Cadaverine

The labeling of actin Gln41 with tetramethylrhodamine cadaverine (TRC, Zedira, Darmstadt, Germany) was performed using bacterial transglutaminase as previously described [34,67]. TRC was covalently attached to Gln41 of actin. For this 24 μM G-actin in G-buffer (0.2 mM CaCl_2_, 0.2 mM ATP, 1 mM DTT and 5 mM Hepes, pH 7.6) was incubated overnight with 36 μM TRC in the presence of 0.04 mg/mL of recombinant bacterial transglutaminase (Zedira). The labeling degree was about 50% as determined by measurement of TRC concentration using molar extinction coefficient 0.078 μM cm^−1^ at 544 nm.

### 4.4. Co-Sedimentation Assay

Binding of Tpm1.1 and Tpm3.12 to 3 µM F-actin was analyzed in 5 mM imidazole, pH 7.5, 1 mM DTT, 100 mM NaCl, and 2 mM MgCl_2_. Tropomyosin was added at concentrations from 0 to 2.5 μM and allowed to bind to F-actin by incubation at room temperature for 20 min. Cofilin-2 affinity for F-actin alone and F-actin saturated with either 1 µM Tpm1.1 or 1 µM Tpm3.12 was analyzed at cofilin concentrations from 0 to 10 µM. The proteins were mixed and incubated for 20 min at room temperature. Co-sedimentation of tropomyosin and cofilin with F-actin was performed as described before [33,68]. Normalization of the obtained tropomyosin/actin and cofilin/actin densitometric ratios was done by dividing the ratios by the maximal value obtained in each experiment. The normalized values were plotted versus the unbound tropomyosin or cofilin concentrations. The apparent association constant (K_app_) and Hill cooperativity coefficient (α^H^) were calculated by fitting the experimental data to Hill equation Equation (1) in SigmaPlot 12.5 (Systat Software Inc.).
(1)$$v=n{\left[free\;L\right]}^{\alpha H}{Kapp}^{\alpha H}/1+{\left[free\;L\right]}^{\alpha H}{Kapp}^{\alpha H}$$
where v is the fractional saturation of F-actin with either tropomyosin or cofilin, n is the maximal saturation of the filament with tropomyosin or cofilin, and [free L] is the concentration of unbound ligand (tropomyosin or cofilin) left in the supernatant. The average values and standard errors were computed in SigmaPlot. Statistical significance of differences between groups was determined by one-way ANOVA.

Binding of cofilin-2 was analyzed according to the McGhee and von Hippel linear lattice model [48] by fitting the experimental points to Equation (2):(2)$$\frac{v}{c}=K_{o}\left(1-n\cdot v\right){\left(\frac{\left(2\omega -1\right)\left(1-n\cdot v\right)+v-R}{2\left(\omega -1\right)\left(1-n\cdot v\right)}\right)}^{\left(n-1\right)}\times {\left(\frac{1-\left(n+1\right)\cdot v+R}{2\left(1-n\cdot v\right)}\right)}^{2}$$
where R = {[1 − (n + 1) υ] 2 + 4ωυ(1 − nυ)}1/2, υ is the number of moles of cofilin-2 bound per mole of actin, n is the stoichiometry of cofilin-2 binding to actin, and c is the concentration of free cofilin.

The gels used for determination of cofilin-2 actin binding in the presence of tropomyosin were also used to analyze cofilin-induced dissociation of tropomyosin from the filament. Tropomyosin/actin densitometric ratios obtained at each cofilin-2 concentration were divided by the maximal value in each experiment and were drawn versus cofilin-2/actin molar ratio. The experimental points were fit to the modified Hill equation Equation (3) in SigmaPlot 12.5 (Systat Software Inc., San Jose, CA, USA).
(3)$$v=n{\left[Cof\right]}^{-\alpha H}k^{-\alpha H}/\left(1+{\left[Cof\right]}^{-\alpha H}k^{-\alpha H}\right)$$
where v is the fraction of tropomyosin bound to actin, [Cof] is the cofilin/actin molar ratio, n is the maximal saturation of the filament with tropomyosin, and k is the cofilin/actin ratio at 50% of tropomyosin dissociation from F-actin. The average values and standard errors of the dissociation parameters were from the statistical data reported by SigmaPlot for each curve.

### 4.5. In Vitro Severing/Depolymerization Assay

Severing and depolymerization rates of TRC-F-actin by cofilin-2 were directly observed in the Nikon Eclipse E600 fluorescence microscope equipped with a color camera. One hour before microscopic observation TRC-labelled F-actin was obtained by dilution of TRC-G-actin with AB buffer (20 mM KCl, 10 mM DTT, 2 mM MgCl_2_, 1 mM EGTA, 10 mM Hepes, pH 7.5) to 320 nM final concentration. When present, tropomyosin isoforms were added to 100 nM final concentration.

In vitro severing/depolymerization assays were performed similarly to the previously described [33]. The flow cells were formed by a coverslip taped to glass slides with double-sided Scotch tape. The glass slide was covered by Sigmacote film (SigmaAldrich, St. Louis, MO, USA). First, the AB buffer (40 µL) and HMM (20 μL, 3.5 μM) were applied to the cell. After 2 min incubation, the flow cell was washed with a blocking buffer (AB buffer containing 2 mg/mL BSA) and then TRC-labeled thin filaments were added. The unbound filaments were washed off with AB buffer supplemented with 3 mg/mL glucose, 0.1 mg/mL glucose oxidase, 0.01 mg/mL catalase. The severing/depolymerization was started by addition of 5 μL of 320 nM cofilin-2 to the flow cells. Images were collected for 90 s, snapshots of the images were taken at 30 s intervals. Severing activity was quantitated by counting the number of breaks produced by cofilin-2 in one filament. Depolymerization was estimated by measuring the length of the longest fragment obtained after severing, using LUCIA G Version 4.61 software.

### 4.6. Tropomyosin Digestion with Trypsin

For the digestion with trypsin (SigmaAldrich, St. Louis, MO, USA), F-Mg-actin was prepared by incubation of the Ca^2+^-binding G-actin for 10 min with 0.2 mM EGTA and 0.5 mM MgCl_2_, followed by polymerization with 100 mM NaCl and 2 mM MgCl_2_ for 30 min at room temperature. Tropomyosin alone (2 µM) and bound to F-Mg-actin (5 µM) was digested with trypsin at 0.02 mg/mL in 5 mM Hepes (pH 7.6), 100 mM NaCl, 5 mM MgCl_2_, 0.2 mM ATP at room temperature. The reaction was terminated after 0–60 min of digestion by 2-fold molar excess of soy-bean trypsin inhibitor (SBTI, SigmaAldrich, St. Louis, MO, USA) over trypsin. Samples were separated on 12% SDS-PAGE gels. Density of tropomyosin bands was quantitated using the EasyDens software. Raw data were normalized using the equation:(4)$$D=\left(D_{t}\times 100\%\right)/D_{t0}$$
where D = normalized density, D_t_ = density at a given time, D_t0_ = density at time 0.

Normalized data were drawn versus time of digestion. Experimental errors were calculated in Excel as standard errors of the mean.

## Figures and Tables

**Figure 1 ijms-21-04285-f001:**
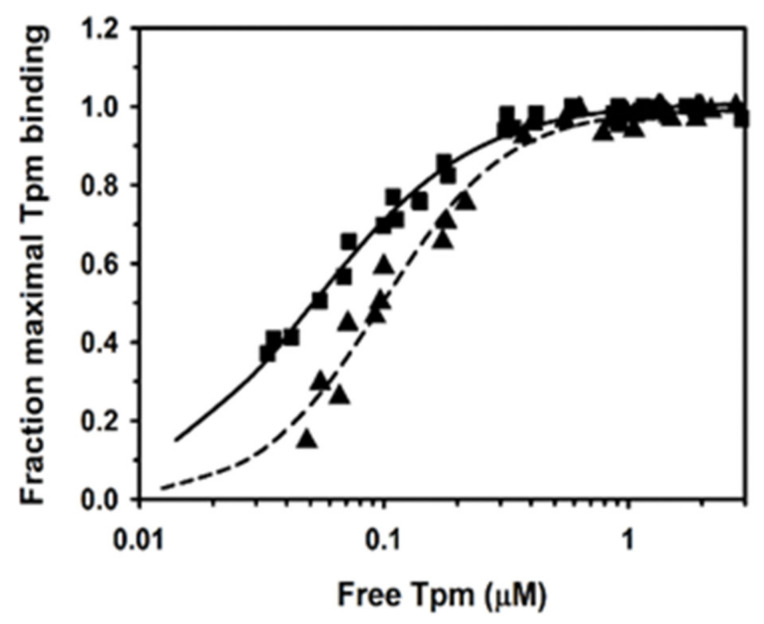
Binding of tropomyosin isoforms to F-actin. Tpm1.1 (triangles, dashed line), Tpm3.12 (squares, solid line) binding to 3 µM F-actin in 100 mM NaCl, 2.0 mM MgCl_2_, 1 mM DTT, 5.0 mM Tris–HCl, pH 7.5. The points are from three experiments. The lines show fitting of the experimental points to Equation (1).

**Figure 2 ijms-21-04285-f002:**
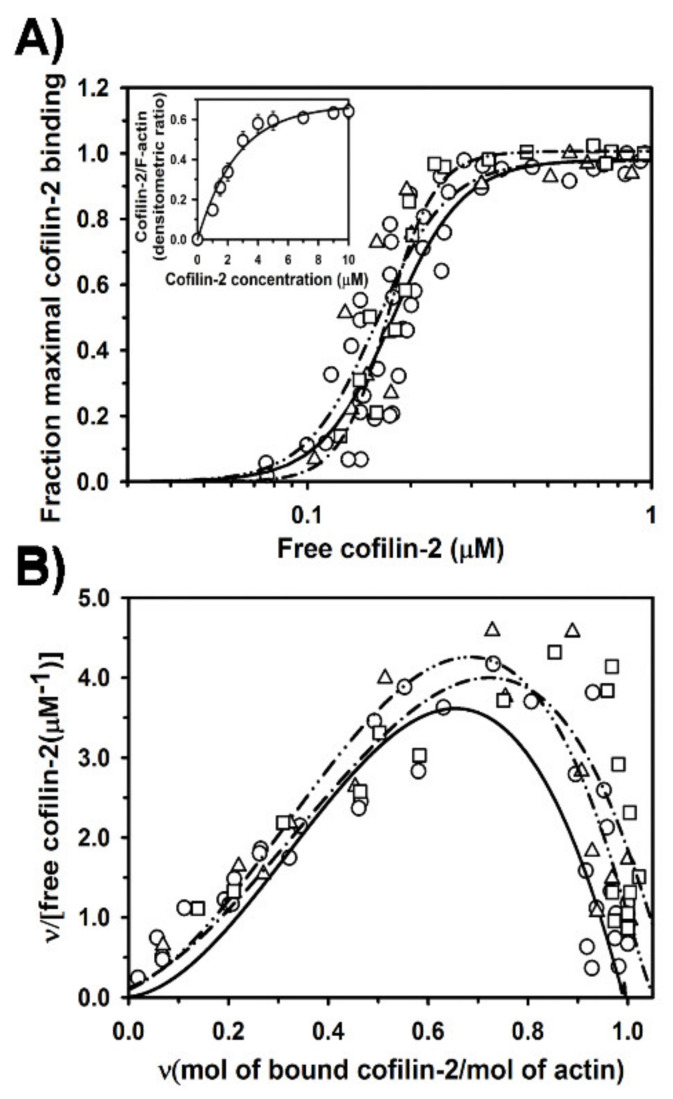
Binding of cofilin-2 to F-actin alone and to F-actin initially saturated with tropomyosin. The experimental points fit to the Hill model (**A**) and the McGhee and von Hippel linear lattice model (**B**). Binding of cofilin-2 to F-actin alone (open circles, solid line), F-actin covered by Tpm1.1 (open triangles, dashed-dot-dot line) or by Tpm3.12 (open squares, dash-dot line). Protein concentration: 3 µM F-actin, 1 µM Tpm1.1 or Tpm3.12, 0–10 µM cofilin-2. Buffer: 100 mM NaCl, 2.0 mM MgCl_2_, 1 mM DTT, 5.0 mM Tris–HCl, pH 7.5. The points are from three experiments. Inset in panel (**A**) illustrates saturation of F-actin alone as a function of total concentration of cofilin-2.

**Figure 3 ijms-21-04285-f003:**
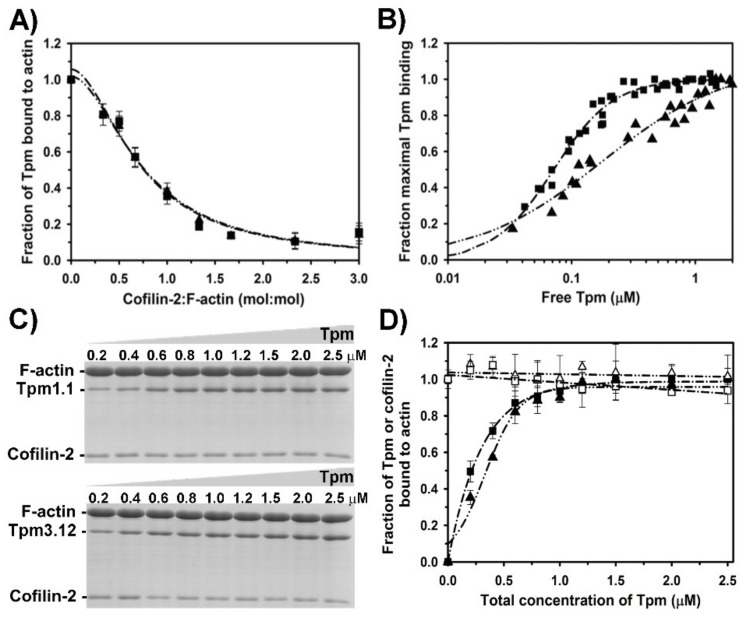
Interactions of Tpm1.1 and Tpm3.12 with F-actin in the presence of cofilin-2. (**A**) Dissociation of Tpm1.1 (triangles, dash-dot-dot line), Tpm3.12 (squares, dash-dot line) from F-actin by increasing concentrations of cofilin-2. The points were averaged from three independent experiments ± SE. Protein concentrations: 3 µM F-actin, 1.0 μM Tpm1.1 or Tpm3.12, 0–10 µM cofilin-2. (**B**) Binding of Tpm1.1 (triangles, dash-dot-dot line), Tpm3.12 (squares, dash-dotted line) to 3 µM F-actin in the presence of 1 µM cofilin-2. The points were collected from three independent experiments. (**C**) Electrophoretic separation of the proteins collected in pellets after ultracentrifugation of 3 μM F-actin with 0–2.5 μM Tpm1.1 (upper panel) or Tpm3.12 (lower panel) in the presence of 1 µM cofilin-2. (**D**) Densitometric analysis of the protein bands separated on gels illustrated in (**C**). Fraction of cofilin-2 bound to F-actin in the presence of either Tpm1.1 (open triangles, dash-dot-dot line) or Tpm3.12 (open squares, dash-dotted line). Fractional saturation of F-actin with Tpm1.1 (triangles, dash-dot-dot line) or Tpm3.12 (squares, dash-dotted line) in the presence of 1 µM cofilin-2. Other conditions: 5.0 mM Tris-HCl, pH 7.5, 100 mM NaCl, 2.0 mM MgCl_2_, 1 mM DTT.

**Figure 4 ijms-21-04285-f004:**
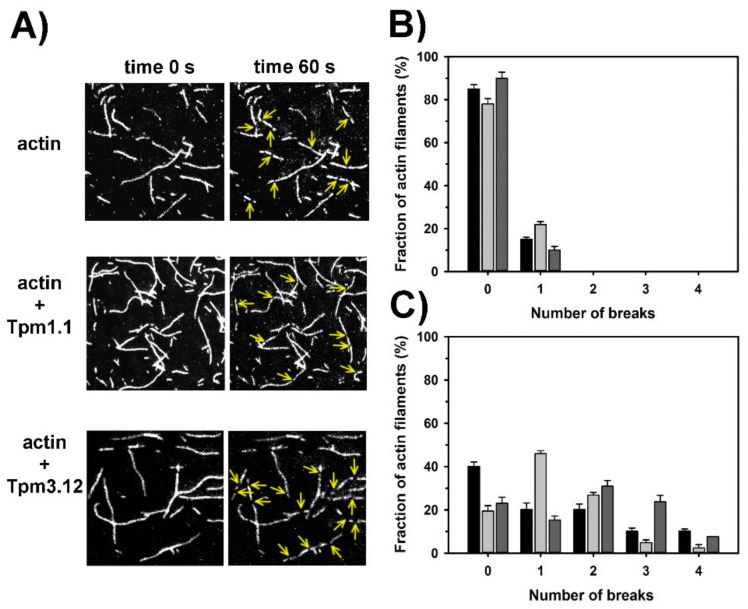
Regulation of severing activity of cofilin-2 by tropomyosin Tpm1.1 and Tpm3.12. (**A**) Snapshots of F-actin alone and F-actin in the presence of tropomyosin isoforms before (time 0 s) and after (60 s) addition of cofilin-2; selected breaks in the filaments are marked with yellow arrows. (**B**) Fraction of filaments with breaks in the absence of cofilin-2. (**C**) Fraction of filaments with breaks 60 s after addition of cofilin-2. The bars are color-coded as follows: F-actin alone (black), F-actin in the presence of Tpm1.1 (light grey), Tpm3.12 (dark grey). Buffer: 20 mM KCl, 2 mM MgCl_2_, 1 mM EGTA, 10 mM DTT, 10 mM Hepes, pH 7.5, supplemented with 0.1 mg/mL glucose oxidase, 0.01 mg/mL catalase, and 3 mg/mL glucose. The experimental errors were SE calculated in Excel (Microsoft Office Professional Plus 2013) from 100 filaments. Analysis of statistical significance of differences between the experimental groups in (B) and (C) (F-actin, F-actin-Tpm1.1, and F-actin-Tpm3.12) showed that the differences for each group were significant (*p* < 0.05).

**Figure 5 ijms-21-04285-f005:**
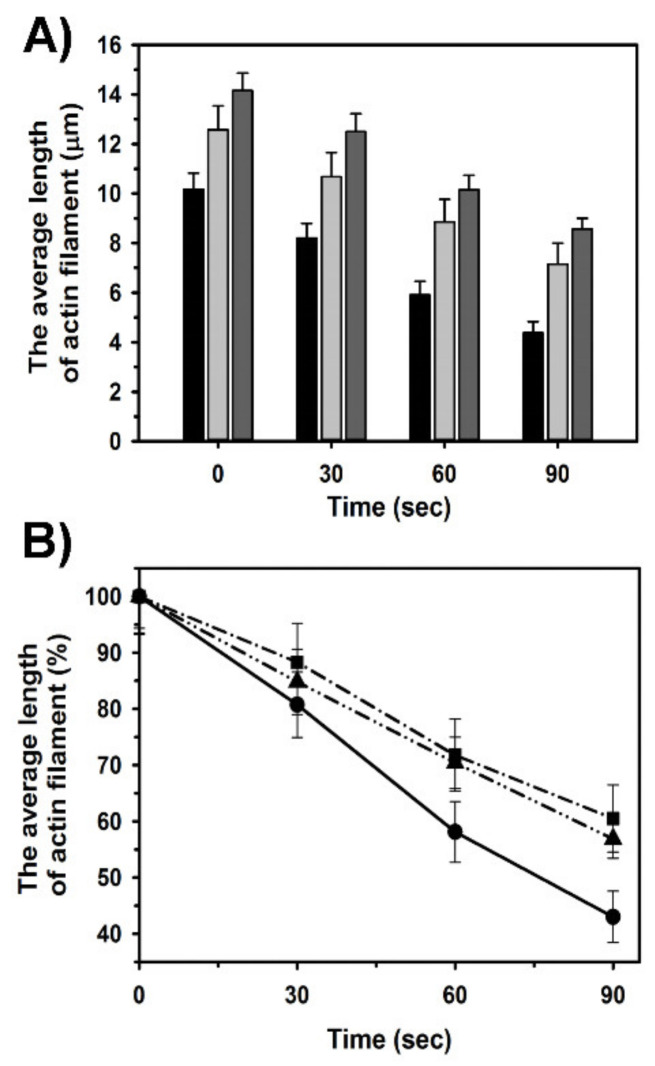
Tropomyosin-dependent regulation of actin filaments shortening by cofilin-2. (**A**) Averaged lengths of the filaments before and after exposition to cofilin-2. F-actin alone (black), F-actin in the presence of Tpm1.1 (light grey), Tpm3.12 (dark grey). (**B**) The rate of the actin filament depolymerization. The lengths of the filaments are shown as % of the lengths obtained before addition of cofilin-2. F-actin alone (circles, solid line), F-actin in the presence of Tpm1.1 (triangles, dash-dot-dot line), Tpm3.12 (squares, dash-dotted line). Conditions as in Figure 4 legend. The experimental errors are SE calculated in Excel (Microsoft Office Professional Plus 2013) from the lengths of 100 filaments. Analysis of statistical significance of differences between F-actin, F-actin-Tpm1.1, and F-actin-Tpm3.12 in (A) showed that the differences were significant (*p* < 0.05); in (**B**) the differences were statistically significant between F-actin and F-actin-Tpm1.1 or F-actin-Tpm3.12 at time points 60 and 90 sec (*p* < 0.001) and not significant at 30 sec (*p* = 0.259); the difference between F-actin-Tpm1.1. and Tpm3.12 was not significant within the whole time range (*p* > 0.10).

**Figure 6 ijms-21-04285-f006:**
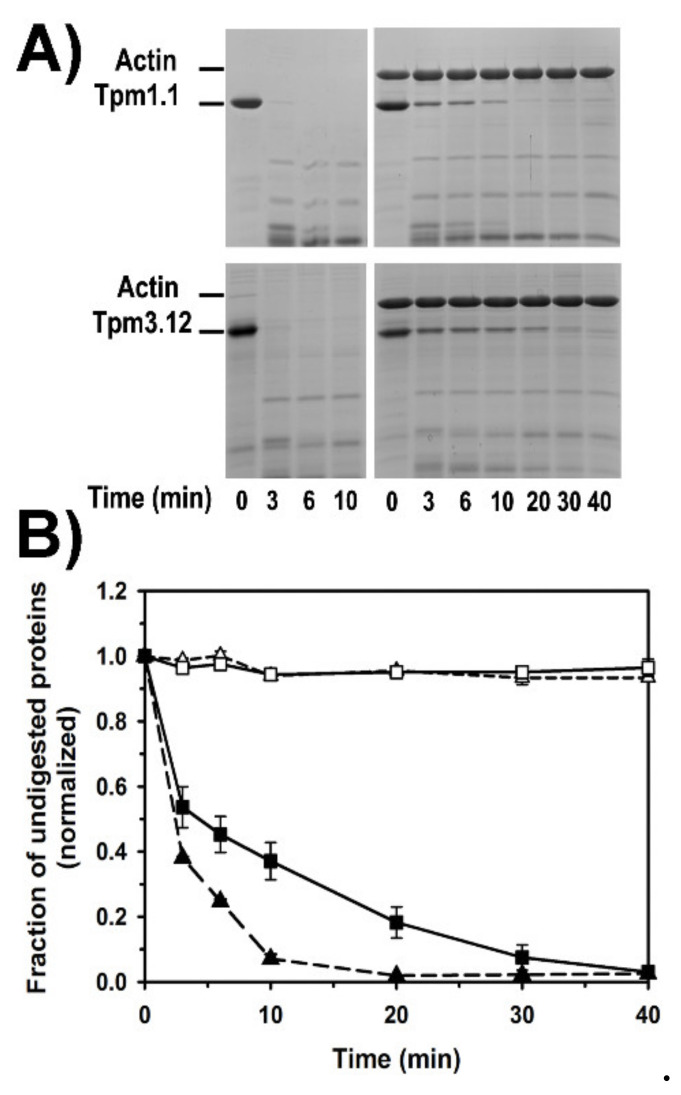
Tryptic digestion of tropomyosin alone and bound to F-actin. (**A**) Representative polyacrylamide gels showing the digestion time course. (**B**) Fraction of undigested F-actin in the presence of Tpm1.1 (open triangles, dashed line), F-actin in the presence of Tpm3.12 (open squares, solid line), Tpm 1.1 in complex with F-actin (black triangles, dashed line), Tpm3.12 in complex with F-actin (black squares, solid line). Experimental conditions: 5 μM F-actin, 2 μM Tpm1.1 or Tpm3.12 in 5 mM Hepes, 100 mM NaCl, 2 mM MgCl_2_, 0.2 mM ATP, pH 7.6. Trypsin was at 1:50 *w*/*w* actin ratio. Experimental points are averages from three independent experiments ± SE. Analysis of statistical significance of differences between the experimental groups in (**B**) (F-actin, F-actin-Tpm1.1, and F-actin-Tpm3.12) showed that the differences were significant at time range from 3 to 20 min (*p* < 0.05); at 30 and 40 min the differences were not significant (*p* > 0.22).

**Figure 7 ijms-21-04285-f007:**
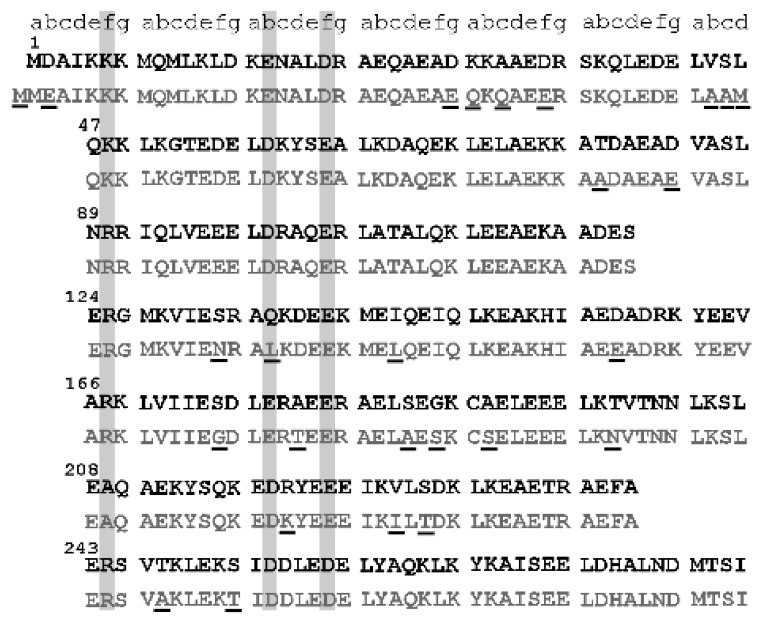
Comparison of amino acid sequences of Tpm1.1 (black) and Tpm3.12 (gray). Amino acid residues corresponding to heptapeptide repeat of the coiled coil (a–g) are shown on top of the sequences. The consensus sites which potentially interact with residues exposed on actin surface [46] are marked with gray boxes. Amino acid residues, which distinguish Tpm3.12 from Tpm1.1 are underlined.

**Table 1 ijms-21-04285-t001:** Parameters of tropomyosin binding to F-actin in the presence and absence of cofilin-2.

	− Cofilin-2		+ Cofilin-2	
	K_app_ (µM^−1^)	α^H^	K_app_ (µM^−1^)	α^H^
Tpm1.1	10.2 ± 0.9 ^a,b^	1.7 ± 0.3 ^b^	5.6 ± 0.9 ^a,b^	0.9 ± 0.1 ^a,b^
Tpm3.12	19.1 ± 1.6 ^a,b^	1.3 ± 0.2	13.9 ± 0.8 ^a,b^	1.8 ± 0.2 ^a^

The parameters were obtained by fitting the experimental data to Equation (1). All parameters are averages ± SE. Protein concentrations: 3 µM F-actin, 1.0 μM cofilin-2, 0–2.5 µM Tpm1.1 or Tpm3.12. ^a^ Statistically significant differences in K_app_ and α^H^ values between Tpm1.1 and Tpm3.12 (*p* < 0.05). ^b^ Statistically significant differences in K_app_ and α^H^ values obtained in the presence and absence of cofilin-2 (*p* < 0.05).

**Table 2 ijms-21-04285-t002:** Parameters of cofilin-2 binding to F-actin in the presence and absence of tropomyosin isoforms.

	K_app_ (µM^−1^)	Κ_0_ (µM^−1^)	ω
Cofilin-2	5.7 ± 0.2	0.01	50
Cofilin-2 + Tpm1.1	6.3 ± 0.4	0.17	12
Cofilin-2 + Tpm3.12	5.8 ± 0.2	0.21	10

K_app_ was computed by fitting the experimental data to Equation (1), K_0_ and ω were obtained by fitting the experimental data to Equation (2). Conditions as specified in Figure 2 legend.

**Table 3 ijms-21-04285-t003:** Parameters of cofilin-induced dissociation of tropomyosin from F-actin.

	50% Dissociation of Tropomyosin (Cofilin-2/Actin)	α^H^
Tpm1.1	0.75 ± 0.04	1.9 ± 0.3
Tpm3.12	0.71 ± 0.03	1.9 ± 0.5

The parameters were obtained by fitting the dissociation curves (Figure 3A) to Equation (3). All parameters shown are averages ± SE. The differences in the half-dissociation parameter were statistically not significant (*p* > 0.05).

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
