# Peer review of "Regulation of Actin Filament Length by Muscle Isoforms of Tropomyosin and Cofilin"

_ijms, 2020, doi:10.3390/ijms21124285_

Round 1

Reviewer 1 Report

The authors have investigated the molecular basis of actin filament length control  by studying two tropomyosin isoforms from muscles with different  filament lengths and their interaction with coffilin 2.  This is an interesting study that indicates that tropomyosin variants play a role in determining actin filament length.   I have a few comments for improvement.

1    Fig 2. Some explanation of the meaning of the Von Hippel plot would help since this may be unfamiliar to many readers.  On first glance 2B is in the form  of a Scatchard plot; what is the meaning of the downturn of the line at high v?

2   The discussion is quite long;  the authors should include subheadings to orientate the reader

3   The paragraph starting at line343 seems to repeat quite a lot of what has already written and should be more concisely written 

4  The comment on the role of tropomodulin at the end of the discussion are interesting.  Could these results be included in this manuscript, please.

Author Response

Answers to the Reviewer’s 1 comments

The authors have investigated the molecular basis of actin filament length control  by studying two tropomyosin isoforms from muscles with different  filament lengths and their interaction with coffilin 2. This is an interesting study that indicates that tropomyosin variants play a role in determining actin filament length. I have a few comments for improvement.

Answer: Thank you for your comments, which helped us to improve the manuscript. The corrections we have introduced are marked in green.

1    Fig 2. Some explanation of the meaning of the Von Hippel plot would help since this may be unfamiliar to many readers. On first glance 2B is in the form of a Scatchard plot; what is the meaning of the downturn of the line at high v?

Answer: Indeed, this is a Scatchard plot. As in the Scatchard analysis, the value on the OX axis, which is approached by the binding curve at high v is the maximal occupancy of F-actin by cofilin. Because v is cofilin/actin molar ratio, v=1.0 shows the stoichiometry of cofilin binding to actin. As requested, appropriate explanations were included in the revised manuscript. The corrections are in  lines 129-136 of the revised manuscript.

2   The discussion is quite long;  the authors should include subheadings to orientate the reader.

Answer: As suggested, Discussion was divided into subsections.

3   The paragraph starting at line 343 seems to repeat quite a lot of what has already written and should be more concisely written.

Answer: We did our best to make this paragraph shorter. We removed the first part of the paragraph, because it was not directly connected to the results obtained in this work.

4  The comment on the role of tropomodulin at the end of the discussion are interesting.  Could these results be included in this manuscript, please.

Answer:  While working on the manuscript we also thought about including the results with tropomodulin 1 (Tmod1). We decided against it, as the results are too preliminary. In addition, including Tmod1 into the story described in this manuscript would require additional experiments, for example, the analysis of the effects of Tmod1 on the lengths of the filaments by the fluorescence microscopy assay. Therefore we decided to remove the sentence which refers to the Tmod1 data from the manuscript. Instead we added an information that more studies are required (line 423-424) to share important questions with the scientific community.

Because this issue evoked the Reviewer’s interest, here we would like to explain our preliminary results. We compared effects of Tpm3.12, two myopathy mutants of Tpm3.12 (A4V and R91C), and Tpm1.1 on the inhibition of the actin pointed end polymerization by Tmod1. The results for Tpm3.12 and the mutants are published (Moraczewska et al., 2019, FEBS J. 286(10):1877-1893. doi: 10.1111/febs.14787). Tpm1.1 was included in those experiments only partially. We wanted to  check whether Tpm1.1 and Tpm3.12 differently regulate elongation at the pointed end and inhibition of this process by Tmod1 . At one Tmod1 concentration we did not observe significant differences between tropomyosin isoforms, hence the remark in the first version of the submitted manuscript. In order to fully confirm this result we have to perform the pointed end polymerization assay at a wide range of Tmod1 concentrations.

Reviewer 2 Report

The homogeneity of the lengths of thin filaments in muscle fibers is truly striking, and it is an important question how the lengths are regulated in a manner depending on the muscle types. In this manuscript, Robaszkiewicz et al. characterized the differential impacts two isoforms of tropomyosin (Tpm) exert on the severing/depolymerizing activity of cofilin against actin filaments. Of the two Tpm isoforms, Tpm1.1 is specific to fast muscle, and Tpm3.12 is specific to slow muscles, respectively. Authors found that Tpm3.12 generated longer actin filaments than Tpm1.1 when cofilin acted on actin filaments, which is interesting in light of the fact that thin filaments in slow muscles are longer than those in fast muscles. Authors further concluded that the differential effects of the Tpm isoforms on filament lengths are due to stronger protective effect of Tpm3.12 on actin filaments against the depolymerizing activity of cofilin. However, in my opinion, the experimental evidence provided here do not convincingly support the conclusion.

Major criticism

  1. The most serious drawback of this paper is the lack of direct evidence that the two isoforms differentially inhibit cofilin-induced depolymerization of actin filaments. This conclusion is based on the observation that the actin filaments bound with Tpm3.12 were longer than those bound with Tpm1.1 when cofilin acted on actin filaments for 30-90 min (Fig 5), even though Tpm3.12 allowed more frequent severing by cofilin than Tpm1.1. A natural inference from those two observations is that Tpm3.12 inhibits cofilin-induced depolymerization more efficiently than Tpm1.1. Although this is a plausible inference, it is nonetheless only an inference. Authors have the time-lapse fluorescence microscopic images of individual actin filaments during the incubation with cofilin in the absence and presence of the Tpm isoforms. This would allow them to perform additional analysis to directly measure the depolymerization speed, such as quantitative kymographic analysis of individual filaments protected either by Tpm1.1 or 3.12, or without Tpms. I think that sort of data would significantly improve the quality of this paper.

  1. Also related to Figure 5. Authors should provide plausible explanation as to why Tpm-bound filaments were longer than plain actin filaments at t=0 when cofilin was added (i.e., before cofilin started to sever or depolymerize the filaments). Is it due to inhibition of spontaneous depolymerization after dilution of the stock solution of actin filaments? Whatever the reason is, the fact that Tpm3.12-bound filaments were longer than those bound with Tpm1.1 to begin with makes it difficult to evaluate the statement that Tpm3.12-bound filaments were truly slower to shrink than those bound with Tpm1.1. Some sort of normalization to correct the initial difference in filament lengths seems necessary.

  1. Some sentences are inappropriately too conclusive. They should be rephrased.

(i) “…decreased cooperativity of cofilin binding should increase severing. Our results demonstrated that this indeed was the case” (line 322-324). For the first sentence, I would use “might” in place of “should”, when it is only a spculation. Regarding the second sentence, there is no direct evidence for this, as explained above.

(ii) “we … demonstrated that muscle isoforms of tropomyosins are able to differentiate length of the filaments, which is important for maintaining the filament length characteristic for fast and slow muscle fibers.” (line 345-346). There is a large gap between the length distribution of actin filaments in a simple in vitro assay and that in muscle fibers. Much more work is needed to justify this conclusive statement.

(iii) “The results presented in this work show that the muscle-type specific tropomyosins are directly involved in this function (=regulation of thin filament length in muscles).” (line 362-262). Again, I believe that the research is too premature to make this conclusion.

Minor comments.

  1. “In striated muscle the length of actin and myosin filaments contributes to the development of force.” (line 13-14). I do not think that length of the filaments contributes to force generation.
  2. A number of sentences that contain the term “saturating” or “saturation” do not make sense to me. They include:

“this analysis was performed to find the concentration of tropomyosin and actin at saturation under salt concentration, which were close to physiological” (line 95-97).

“Tpm1.1, Tpm3.12 saturating 3 µM to F-actin in 5.0 mM Tris-HCl …” (line 105-106).

“at 1 µM concentration cofilin-2 saturated 3 µM F-actin in about 25%. At this concentration of cofilin-2 the filament was saturated with tropomyosin in about 80%” (line 166-168)

  1. Please specify the concentration of cofilin 2 in Table 1.
  2. Table 1 (line 128) -> Table 2.
  3. Inset in Figure 2A is too small.
  4. “lack of significant of tropomyosin” (line 172) ??
  5. “increasing concentration of tropomyosin did not remove cofilin-2 from the filament (Figure 3C)” (line 173-174). To me, it appears that increasing concentration of Tpm3.12 removed cofilin-2 from the filament.
  6. “addition of cofilin-2 at 1:1 molar ratio” (line 191). 1:1 molar ratio to what? If it is actin, how did Authors determine the concentration of actin in the flow cell? Note that the actin concentration in the flow cell is certainly much lower than that in the solution that was introduced into the flow cell.
  7. Why 20% filaments had a break even before the addition of cofilin (Fig 4B)? Is it not due to the residual motor activity of HMM, activated by trace amount of ATP carried over from G-buffer? This is probably not a fatal problem, but I do not understand why Authors complicated the experiment by introducing a third actin binding protein, which is reported to affect the actin structure, when they are trying to investigate the relationship between two actin binding proteins…
  8. Lower left fluorescence image in Figure 4 is odd. The horizontal filaments are much thicker than the vertical ones. Why?
  9. “muscle isoform of cofilin was shown to maintain the length of sarcomeric actin filaments” (line 260-261). -> “muscle isoform of cofilin was shown to participate in maintenance of the length of sarcomeric actin filaments”

Author Response

Answers to the Reviewer’s 2 comments

The homogeneity of the lengths of thin filaments in muscle fibers is truly striking, and it is an important question how the lengths are regulated in a manner depending on the muscle types. In this manuscript, Robaszkiewicz et al. characterized the differential impacts two isoforms of tropomyosin (Tpm) exert on the severing/depolymerizing activity of cofilin against actin filaments. Of the two Tpm isoforms, Tpm1.1 is specific to fast muscle, and Tpm3.12 is specific to slow muscles, respectively. Authors found that Tpm3.12 generated longer actin filaments than Tpm1.1 when cofilin acted on actin filaments, which is interesting in light of the fact that thin filaments in slow muscles are longer than those in fast muscles. Authors further concluded that the differential effects of the Tpm isoforms on filament lengths are due to stronger protective effect of Tpm3.12 on actin filaments against the depolymerizing activity of cofilin. However, in my opinion, the experimental evidence provided here do not convincingly support the conclusion.

Answer: Thank you for reading the manuscript carefully and suggesting improvements. The answers to your critical remarks are given below, the changes we’ve have made are marked in purple.

Major criticism

  1. The most serious drawback of this paper is the lack of direct evidence that the two isoforms differentially inhibit cofilin-induced depolymerization of actin filaments. This conclusion is based on the observation that the actin filaments bound with Tpm3.12 were longer than those bound with Tpm1.1 when cofilin acted on actin filaments for 30-90 min (Fig 5), even though Tpm3.12 allowed more frequent severing by cofilin than Tpm1.1. A natural inference from those two observations is that Tpm3.12 inhibits cofilin-induced depolymerization more efficiently than Tpm1.1. Although this is a plausible inference, it is nonetheless only an inference. Authors have the time-lapse fluorescence microscopic images of individual actin filaments during the incubation with cofilin in the absence and presence of the Tpm isoforms. This would allow them to perform additional analysis to directly measure the depolymerization speed, such as quantitative kymographic analysis of individual filaments protected either by Tpm1.1 or 3.12, or without Tpms. I think that sort of data would significantly improve the quality of this paper.

Answer: The normalization of the depolymerization rates, which was added as Figure 5B, shows that the two isoforms do not differentiate the depolymerization rate. The difference is in the stabilization of the filaments by maintaining different filament length. The text was revised and the conclusions regarding differential inhibition of cofilin-induced depolymerization of actin filaments by tropomyosin isoforms were corrected in:

line 18 “and depolymerization”  - removed

line 243 the statement “In the presence of both tropomyosin isoforms the rate of the filament shortening was similar” was added.

line 252 “and to slow down depolymerization induced by cofilin-2” – removed

We did not attempt to quantitate the filaments by kymographic analysis, because measurements of an individual filament depolymerization rate is measured from one end of the filament while the other is anchored to the coated coverslip (e.g.  Shekhar & Carlier, 2017, Current Biology 27, 1990–1998, 2017). This is done with the use of microfluidics coupled to a  high-class TIRF microscope, which allows to obtain high-quality actin filament images. We could not do the analysis, because our filaments were immobilized not at one end, but along the entire length. The quality of the images obtained in the epifluorescence microscope makes it difficult or even impossible to perform precise kymographic analyzes.

  1. Also related to Figure 5. Authors should provide plausible explanation as to why Tpm-bound filaments were longer than plain actin filaments at t=0 when cofilin was added (i.e., before cofilin started to sever or depolymerize the filaments). Is it due to inhibition of spontaneous depolymerization after dilution of the stock solution of actin filaments? Whatever the reason is, the fact that Tpm3.12-bound filaments were longer than those bound with Tpm1.1 to begin with makes it difficult to evaluate the statement that Tpm3.12-bound filaments were truly slower to shrink than those bound with Tpm1.1. Some sort of normalization to correct the initial difference in filament lengths seems necessary.

Answer: Different rates of spontaneous depolymerization after dilution of the filaments is one plausible explanation. The second possibility is that the filaments, which polymerize in the presence of Tpm are longer than the plain actin filaments and they stay longer after dilution. These explanations were included into the Discussion section 3.4 (lines 398-401).

A normalization of the depolymerization rate was added to Figure 5 as panel B. The normalized data show that the rate of depolymerization in the presence of both tropomyosins is the same. It doesn’t change the result shown in panel A showing that in the presence of Tpm3.12 the filaments left after severing are longer. We explain this in section 2.7 by lower dynamics Tpm3.12 binding to actin.

  1. Some sentences are inappropriately too conclusive. They should be rephrased.

(i) “…decreased cooperativity of cofilin binding should increase severing. Our results demonstrated that this indeed was the case” (line 322-324). For the first sentence, I would use “might” in place of “should”, when it is only a spculation. Regarding the second sentence, there is no direct evidence for this, as explained above.

Answer: We have made the suggested change in the first sentence and removed the second one (line 356).

(ii) “we … demonstrated that muscle isoforms of tropomyosins are able to differentiate length of the filaments, which is important for maintaining the filament length characteristic for fast and slow muscle fibers.” (line 345-346). There is a large gap between the length distribution of actin filaments in a simple in vitro assay and that in muscle fibers. Much more work is needed to justify this conclusive statement.

Answer: That is true, the conclusion was too strong. We removed the second part of the sentence (line 395).

(iii) “The results presented in this work show that the muscle-type specific tropomyosins are directly involved in this function (=regulation of thin filament length in muscles).” (lines 362-262). Again, I believe that the research is too premature to make this conclusion.

 Answer: Following the advice of Reviewer 1, the entire paragraph was removed.

Minor comments.

  1. “In striated muscle the length of actin and myosin filaments contributes to the development of force.” (lines 13-14). I do not think that length of the filaments contributes to force generation.

Answer: For clarity we restated the sentence as: " In striated muscle the extent of the overlap between actin and myosin filaments contributes to the development of force.” (lines 13-14)

  1. A number of sentences that contain the term “saturating” or “saturation” do not make sense to me. They include:

“this analysis was performed to find the concentration of tropomyosin and actin at saturation under salt concentration, which were close to physiological” (line 95-97).

“Tpm1.1, Tpm3.12 saturating 3 µM to F-actin in 5.0 mM Tris-HCl …” (line 105-106).

 “at 1 µM concentration cofilin-2 saturated 3 µM F-actin in about 25%. At this concentration of cofilin-2 the filament was saturated with tropomyosin in about 80%” (line 166-168)

Answer: All three sentences were corrected (marked in purple).

  1. Please specify the concentration of cofilin 2 in Table 1.

Answer: Done as requested.

  1. Table 1 (line 128) -> Table 2.

Answer: The mistake was corrected.

  1. Inset in Figure 2A is too small.

Answer: The symbols and letters were enlarged.

  1. “lack of significant of tropomyosin” (line 172) ??

               Answer: For the Reviewer 3 request, this part of the text was changed.

  1. “increasing concentration of tropomyosin did not remove cofilin-2 from the filament (Figure 3C)” (line 173-174). To me, it appears that increasing concentration of Tpm3.12 removed cofilin-2 from the filament.

Answer:  To check if cofilin-2 was partially removed we analyzed density of protein bands. The normalized data are illustrated in Figure 3D, the result is described and marked in turquoise (lines 239-243).

  1. “addition of cofilin-2 at 1:1 molar ratio” (line 191). 1:1 molar ratio to what? If it is actin, how did Authors determine the concentration of actin in the flow cell? Note that the actin concentration in the flow cell is certainly much lower than that in the solution that was introduced into the flow cell.

Answer: We agree. Flow cells were washed with GOC after actin filaments were immobilized, therefore it is impossible to say what are the concentrations of actin and cofilin-2. We removed this part of the sentence (line 211).

  1. Why 20% filaments had a break even before the addition of cofilin (Fig 4B)? Is it not due to the residual motor activity of HMM, activated by trace amount of ATP carried over from G-buffer? This is probably not a fatal problem, but I do not understand why Authors complicated the experiment by introducing a third actin binding protein, which is reported to affect the actin structure, when they are trying to investigate the relationship between two actin binding proteins…

Answer: Indeed, when the filaments were introduced to the flow cell, traces of ATP could activate motor activity of HMM, which could cause filament breaking. This fact is acknowledged in the revised manuscript (lines 360-365).

We think that our experimental set up serves well the purpose - the comparison of the regulatory effects of tropomyosin isoforms on cofilin-2 severing activity and actin depolymerization. Actin filaments were immobilized by myosin heads bound in rigor. It is true that myosin heads affect actin structure, but the high resolution structure of F-actin-Tpm bound to myosin S1 showed only small myosin-induced conformational changes in actin (von der Ecken et al. Nature 2016). Ngo KX et al. (Scientific Reports 2016) were able to dock cofilin to actin-myosin rigor complex without causing clashes  (Ngo KX et al. Scientific Reports 2016).

  1. Lower left fluorescence image in Figure 4 is odd. The horizontal filaments are much thicker than the vertical ones. Why?

Answer: Thank you for noticing. The filaments were thicker because of excessive image enlargement. The problem was fixed.

  1. “muscle isoform of cofilin was shown to maintain the length of sarcomeric actin filaments” (line 260-261). -> “muscle isoform of cofilin was shown to participate in maintenance of the length of sarcomeric actin filaments”

Answer: Sounds better. Thank you. (lines 289-290)

Reviewer 3 Report

This study investigates the regulation of actin filament length by tropomyosin isoforms, Tpm1.1 and Tpm3.12, and cofilin-2. The experiments are well designed, the results are statistically analyzed, and the discussion is also adequate. I recommend the acceptance of publication of this manuscript after the response of following comments.

  1. In Page 5, line 172-174, the author mentioned “In agreement with the lack of significant of tropomyosin on cofilin-2 affinity for F-actin described above increasing concentrations of tropomyosin did not remove cofilin-2 from the filament (Figure 3C)”. However, it could be seen that the intensity of cofilin band in the both gel images were gradually decreased with increasing of tropomyosin concentration. The author should quantify the intensity of cofilin-2 band. For example, the ratio of cofilin to F-actin (percentage of cofilin-2/F-actin).

  1. In Fig. 4, F-actin is immobilized using HMM. As introduced in the introduction section of this paper, tropomyosin affects actin-myosin interaction. Myosin also affects the binding of cofilin to F-actin (https://doi.org/10.1038/srep35449). The author should mention the effects of immobilization with myosin in the paper.

  1. In page 7, line 215-219, the author demonstrated that both tropomyosin isoforms increased length of F-actin compare to control sample and that the tropomyosin decorated F-actin were remained longer than control sample (Figure 5). If the author might assert the important of each tropomyosin isoform on "slowdown of F-actin depolymerization induced by cofilin-2 (page 8, line 229)", decreased ratio of F-actin length (percentage to 0 sec length) should be showed in Figure 5.

Minor comment

Page3, Line128, isn't "Table 1" a mistake for "Table 2"?

Author Response

Answers to the Reviewer’s 3 comments

This study investigates the regulation of actin filament length by tropomyosin isoforms, Tpm1.1 and Tpm3.12, and cofilin-2. The experiments are well designed, the results are statistically analyzed, and the discussion is also adequate. I recommend the acceptance of publication of this manuscript after the response of following comments.

Answer: Thank you for your kind opinion and helpful suggestions. The responses to your comments are below. All changes in the manuscript are marked in turquoise.

1) In Page 5, line 172-174, the author mentioned “In agreement with the lack of significant of tropomyosin on cofilin-2 affinity for F-actin described above increasing concentrations of tropomyosin did not remove cofilin-2 from the filament (Figure 3C)”. However, it could be seen that the intensity of cofilin band in the both gel images were gradually decreased with increasing of tropomyosin concentration. The author should quantify the intensity of cofilin-2 band. For example, the ratio of cofilin to F-actin (percentage of cofilin-2/F-actin).

Answer: Following this advice, we analyzed the densitometric ratio of cofilin-2 to F-actin covered by Tpm1.1 or Tpm3.12 found in pellets. The line fitted to the normalized experimental points has a slight downward trend, which was sensed by the Reviewer. However, the decrease in the occupancy of the filament by cofilin is very small. Even 5-fold molar excess of tropomyosin did not remove cofilin-2 from the filaments. To illustrate this result we included the results of densitometric analysis as Figure 3D.

 2) In Fig. 4, F-actin is immobilized using HMM. As introduced in the introduction section of this paper, tropomyosin affects actin-myosin interaction. Myosin also affects the binding of cofilin to F-actin (https://doi.org/10.1038/srep35449). The author should mention the effects of immobilization with myosin in the paper.

Answer: In the Introduction section we mentioned the role of tropomyosin in the Ca-dependent control of actin-myosin interactions in the presence of troponin complex. In the fluorescence microscopy assay we did not use troponin, so that in the absence of ATP myosin shifted tropomyosin away from the actin-myosin binding sites. The mutually exclusive binding of myosin and cofilin, which led to clustering of cofilin on the filament, was shown by Ngo KX et al. (Scientific Reports 2016) in the presence of low concentrations of ATP. As pointed by the authors, low ATP concentrations were necessary to reposition myosin heads on the filament. In our experiments myosin heads were bound to actin in rigor and they were attached to the glass surface, therefore the condition of both experiments cannot be compared.

As suggested by the Reviewer, the effects of actin immobilization by myosin heads are discussed in the revised manuscript (lines 366-371).

3) In page 7, line 215-219, the author demonstrated that both tropomyosin isoforms increased length of F-actin compare to control sample and that the tropomyosin decorated F-actin were remained longer than control sample (Figure 5). If the author might assert the important of each tropomyosin isoform on "slowdown of F-actin depolymerization induced by cofilin-2 (page 8, line 229)", decreased ratio of F-actin length (percentage to 0 sec length) should be showed in Figure 5.

Answer: As suggested, we added panel B to Figure 5. The protective role of tropomyosin isoforms was quantitated by dividing the average filament length obtained at each time point by the average values obtained in the absence of cofilin. This analysis showed that both studied tropomyosin isoforms slowed down depolymerization of actin filaments by cofilin-2. The rate of depolymerization of both tropomyosin isoforms was similar. The text has been supplemented with these explanations (lines 239-243).

Minor comment

Page3, Line128, isn't "Table 1" a mistake for "Table 2"?

Answer: Thank you. It was a mistake. On page 4 we refer to Table 2. The typo was corrected (line 136).

Round 2

Reviewer 2 Report

In this revised version, Authors performed an indirect but quantitative comparison of depolymerization rates of actin filaments that were bound with either Tpm1.1 or Tpm3.12 (Fig 5B), and now conclude that the depolymerization rates are not significantly different between the two filament forms. I am not convinced that one needs high quality fluorescence images of individual actin filaments to directly measure the depolymerization rates by kymographic analysis. However, since Authors have now dropped the original claim that the depolymerization rates are different between filaments bound with Tpm3.12 and those bound with Tpm1.1, the issue is no longer relevant. Other minor problems are also properly addressed (but see below), and I do not see no problems in scientific soundness in this revised manuscript.

Authors maintain the claim that Tpm3.12-bound filaments are similarly longer than those bound with Tpm1.1, before and after the addition of cofilin, and they now attribute the difference to processes in polymerization or dilution of the polymerized filaments (line 398-401). I agree with this interpretation. However, those effects are not related to cofilin, so that the original scope of the research, which is to investigate the regulation of filament length by Tpm isoforms in connection with cofilin activities, becomes more or less irrelevant. I think it is a matter of editorial decision to accept or reject such work.

A new minor problem: “polymerization rate” (line 401) -> “depolymerization rate”?

Author Response

Answers to Reviewer 2 comments

In this revised version, Authors performed an indirect but quantitative comparison of depolymerization rates of actin filaments that were bound with either Tpm1.1 or Tpm3.12 (Fig 5B), and now conclude that the depolymerization rates are not significantly different between the two filament forms. I am not convinced that one needs high quality fluorescence images of individual actin filaments to directly measure the depolymerization rates by kymographic analysis. However, since Authors have now dropped the original claim that the depolymerization rates are different between filaments bound with Tpm3.12 and those bound with Tpm1.1, the issue is no longer relevant. Other minor problems are also properly addressed (but see below), and I do not see no problems in scientific soundness in this revised manuscript.

Authors maintain the claim that Tpm3.12-bound filaments are similarly longer than those bound with Tpm1.1, before and after the addition of cofilin, and they now attribute the difference to processes in polymerization or dilution of the polymerized filaments (line 398-401). I agree with this interpretation. However, those effects are not related to cofilin, so that the original scope of the research, which is to investigate the regulation of filament length by Tpm isoforms in connection with cofilin activities, becomes more or less irrelevant. I think it is a matter of editorial decision to accept or reject such work.

Answer: There is some misunderstanding in this matter. When revising the manuscript, we added the explanation suggested by the Reviewer in his/her first review, that the different lengths of the filaments after they were diluted for the microscopy assay may be due to reduced depolymerization rate of the filaments bound to tropomyosin. However, this explanation applies to the filaments’ length before addition of cofilin. When cofilin was added to the flow cell, it started severing and depolymerization. Thus, if cofilin was included in the experiment, the result is related to cofilin. Because there was no difference in the rate of cofilin-induced depolymerization of the filaments associated with either Tpm3.12 or Tpm1.1, the filaments which remained after severing and depolymerization by cofilin had different lengths. This result is related to cofilin activity, because it was obtained in the presence of cofilin actively cutting and depolymerizing the filaments. To make it more clear, we’ve added additional sentences to the discussion (lines 402-405).

The Reviewer focuses on the regulation of the filament length, but we have also shown that both tropomyosins increase severing of the filament by cofilin, which we have attributed to the reduced cooperativity with which cofilin binds to F-actin-Tpm. This result is also related to cofilin activity. Altogether, the results are within the scope of the study aimed at understanding the mechanism of regulation of the actin filament length in slow and fast skeletal muscle by Tpm1.1, Tpm3.12, and cofilin-2. 

A new minor problem: “polymerization rate” (line 401) -> “depolymerization rate”?

Answer: Corrected.